# Implementation of a Sulfide–Air Fuel Cell Coupled to a Sulfate-Reducing Biocathode for Elemental Sulfur Recovery

**DOI:** 10.3390/ijerph18115571

**Published:** 2021-05-23

**Authors:** Enric Blázquez, David Gabriel, Juan Antonio Baeza, Albert Guisasola, Pablo Ledezma, Stefano Freguia

**Affiliations:** 1GENOCOV Research Group, Department of Chemical, Biological and Environmental Engineering, School of Engineering, Universitat Autònoma de Barcelona, 08193 Barcelona, Spain; david.gabriel@uab.cat (D.G.); JuanAntonio.Baeza@uab.cat (J.A.B.); albert.guisasola@uab.cat (A.G.); 2Advanced Water Management Centre, The University of Queensland, Brisbane 4072, Australia; p.ledezma@awmc.uq.edu.au (P.L.); stefano.freguia@unimelb.edu.au (S.F.)

**Keywords:** bio-electrochemical systems (BES), fuel cells (FC), sulfate removal, resource recovery, oxygen-reducing cathode

## Abstract

Bio-electrochemical systems (BES) are a flexible biotechnological platform that can be employed to treat several types of wastewaters and recover valuable products concomitantly. Sulfate-rich wastewaters usually lack an electron donor; for this reason, implementing BES to treat the sulfate and the possibility of recovering the elemental sulfur (S^0^) offers a solution to this kind of wastewater. This study proposes a novel BES configuration that combines bio-electrochemical sulfate reduction in a biocathode with a sulfide–air fuel cell (FC) to recover S^0^. The proposed system achieved high elemental sulfur production rates (up to 386 mg S^0^-S L^−1^ d^−1^) with 65% of the sulfate removed recovered as S^0^ and a 12% lower energy consumption per kg of S^0^ produced (16.50 ± 0.19 kWh kg^−1^ S^0^-S) than a conventional electrochemical S^0^ recovery system.

## 1. Introduction

Bio-electrochemical systems (BES) are a novel technology based on electrochemical processes where the oxidation and/or reduction reactions are catalyzed by microorganisms [1]. BES allow for the treatment of various domestic and industrial wastewaters with concomitant recovery of reusable products such as heavy metals [2,3], nutrients [4,5] and organic compounds [6,7]. BES can also facilitate the treatment of sulfate-rich effluents [8] with the possibility to recover elemental sulfur (S^0^), a product with a stable worldwide demand of about 70 million tons per year just in the fertilizer market [9].

Sulfate-rich wastewaters are produced by several large-scale industrial activities such as pulp and paper production, food processing, dye and detergent manufacturing, among others [10], and if released untreated onto the environment may lead to several environmental problems–e.g., due to the biological formation of sulfide when microorganisms use sulfate as terminal electron acceptor [11]–such as odor nuisance, corrosion and toxicity [12]. Sulfate is the highest oxidation form of sulfur and therefore requires a reduction reaction to be removed and recovered as S^0^. However, there are several wastewaters with high concentrations of sulfate that lack the electron donors required for this reduction [13,14]. The lack of sufficient electron donor causes a need for external electron donor supply, therefore increasing significantly the operational/treatment costs and complexity [15] e.g., over-dosing of electron donor and/or dissolved oxygen can result in the undesirable formation of sulfide, an inefficiency that happens even in the most carefully operated industrial-scale desulfurization systems [16]. BES offer an alternative to this issue by facilitating the autotrophic biological sulfate reduction in a biocathode with electricity as the sole electron donor [17,18,19,20,21]. BES-driven sulfate removal has been previously proven and coupled to anodic sulfide oxidation [22,23,24,25,26,27] to obtain S^0^ as the predominant oxidation product. In addition, complete heterotrophic sulfate reduction with partial sulfide anodic oxidation to sulfur in a single reactor [28,29] at expenses of external organic matter supply, and the treatment of real industrial effluents such as acid mine drainage [30] and flue gas desulfurization effluents [31] have been also demonstrated.

Recently, Blázquez et al. [32] integrated an electrolysis cell (EC) into a BES in order to reduce the energy losses and improve the S^0^ production, in a chamber containing both the biocathode for sulfate reduction and the anode for sulfide oxidation. Good results on elemental sulfur production (up to 93% of sulfate reduced converted into S^0^) were observed when the current of the EC was controlled at 7.5 A m^−2^, but at the expense of low electron recoveries and the requirement of power supply. However, as Dutta et al. [33] previously demonstrated, sulfide can be spontaneously oxidized in the anode of a fuel cell producing elemental sulfur as predominant final oxidation product.

The aim of this study was to integrate a bio-electrochemical cell for autotrophic sulfate reduction with a sulfide–air fuel cell to remove sulfate and recover elemental sulfur in an integrated single electrochemical reactor configuration (referred hereafter as a BES-FC reactor), therefore reducing capital and operational costs–particularly specific energy requirements per kg of sulfate removed and S^0^ produced.

## 2. Experimental

### 2.1. Reactor Description

The BES-FC reactor (Figure 1) was a modification of the reactor used in our previous work [32] with three parallel acrylic frames (internal dimensions of 20 × 5 × 2 cm^3^ each frame) bolted together. The BES consisted of an abiotic anode made of platinum wire (purity 99.95%, 0.50 mm diameter × 50 mm long; Advent Research Materials, Eynsham, UK) located in the first frame (from left to right). A cation-exchange membrane (CMI-7000; Membranes International, USA) was used to separate the biocathode from the anode.

Graphite granules with a diameter of 6 mm or higher (El Carb 100; Graphite Sales, Nova, OH, USA) were used as cathode material, which was placed in the second frame. The third frame contained the fuel cell (FC) that consisted of an anode made of reticulated vitreous carbon (RVC) of 19 × 4 × 1 cm^3^ (45 ppi pore size, Duocel RVC foam; ERG, USA–chosen because it can facilitate spontaneous HS^−^ oxidation without significant S_0_ deposition/fouling [34]), a plastic mesh separator to avoid short-circuiting between the two electrodes of the FC and a passive gas diffusion air-cathode. The latter consisted of a carbon cloth coated with carbon powder and platinum suspension on the inner side (0.125 mg cm^−2^, platinum nominally 40% on high surface area advanced carbon support HiSPEC 4100^TM^ powder; Alfa Aesar, Heysham, UK), whereas the outer side was coated with a polytetrafluoroethylene (PTFE 60% *w*/*w* dispersion in H_2_O; Sigma-Aldrich, St. Luis, MO, USA) solution (4 layers), which permitted oxygen diffusion into the cell while preventing electrolyte permeation [35,36]. One saturated calomel reference electrode (SCE; RE-2BP KCl sat., equiv. +0.244 V vs. Standard Hydrogen Electrode - SHE - at 25 °C; BASi, USA) was embedded in the biocathode frame to measure/control the biocathode potential of the BES. Current density for BES and FC in A m^−2^ was defined as the measured current normalized versus the membrane surface area (100 cm^2^). The pH of the biocathode was controlled by addition of 1 M HCl by a pH controller (Liquisys M CPM253; Endress+Hauser, Reinach, Switzerland).

### 2.2. Biotic and Abiotic Reactions in the BES-FC Process

The BES-FC setup undertakes five simultaneous (bio)electrochemical reactions as described in Equations (1)–(5) below [30,32]. Following Figure 1 from left to right, these reactions occur on the surface of the four electrodes as follows:BES anode (abiotic): 4H_2_O → 2O_2_ + 8H^+^ + 8e^−^(1)
BES-cathode (abiotic): 8H^+^ + 8e^−^ → 4H_2_(2)
(biotic): 4H_2_ + SO_4_^2−^ + H^+^ → HS^−^ + 4H_2_O(3)
FC-anode (abiotic): HS^−^ → S^0^(s) + H^+^ + 2e^−^(4)
FC cathode (abiotic): O_2(g)_ + 4 H^+^ + 4e^−^ → 2H_2_O(5)

### 2.3. Operational Conditions

The system was inoculated with biomass from a previous reactor containing predominantly *Desulfovibrio* sp. [19] and was enriched as described in our previous study, using a cultivation mineral medium without organic carbon for the biocathode at fixed potential (100 mV steps from −0.7 to −0.9 V vs. SHE in 15d intervals as a method of biofilm enrichment based on sulfate removal rates; see rationale and details in [32]). All electrolytes were recirculated within their respective chambers at 125 mL min^−1^ for adequate mixing and minimization of mass-transfer limitations. After the inoculation, the BES-FC was operated in a continuation of five experimental phases with their key operating parameters shown in Table 1.

The FC was switched on in experimental period II (i.e., the electrical circuit was closed between RVC anode and air-cathode from this point onwards). The external resistance used in the FC was 95 Ω, which was the optimum value for maximum power transfer according to the polarization results (Figure 2A). Polarization of the FC was performed with a VMP-3 potentiostat/galvanostat (Bio-Logic, Seyssinet-Pariset, France) from the open circuit voltage to 0 V at 1 mV s^−1^ scan rate. From period II, the current production of the FC was recorded every 60 s using an Agilent 34970A data acquisition unit. Period III started at day 94 with the pH controlled at 7.5 and the cathodic potential of the BES set at −0.95 V vs. SHE to increase the HS proportion up to 70% (according to the pKa of 7 for H_2_S) but maintaining a similar current density. In addition, pH of 7.5 lays in the optimum range (from 6 to 8) of sulfate reducing bacteria (SRB) growth [37]. The air-cathode was replaced for a new one in period III to improve its operation and was characterized by cyclic voltammetry (CV) from −0.2 to +0.5 V vs. SHE at a scan rate of 1 mV s^−1^. In the last experimental period (IV), a flat rubber sheet was placed between the biocathode of the BES and the anode of the FC to mitigate the effects of electron-shuttling compounds and the current density output of the BES was controlled at 10 A m^−2^.

### 2.4. Analytical Methods

All samples were filtered at 0.22 µm (Millipore, Burlington, MA, USA) and diluted using a sulfide anti-oxidant buffer (SAOB) solution in order to minimize sulfide oxidation [38]. Sulfate, sulfite, total dissolved sulfide (TDS) and thiosulfate were measured using an ion chromatograph (IC) with ultra-violet (UV) and conductivity detector (Dionex ICS-2000; Sunnyvale, CA, USA). Polysulfide was assumed to be the difference between the sulfate after H_2_O_2_ oxidation and the sum of all sulfur species measured before H_2_O_2_ oxidation [39]. Elemental sulfur was calculated as the difference between all measured sulfur species of the inlet and all measured sulfur species of the outlet as explained elsewhere [30].

### 2.5. Calculations

The observed sulfate removal rate (SRR, Equation (6)), sulfide production rate (SPR, Equation (7)), theoretical elemental sulfur production rate (TESPR, Equation (8)) and the elemental sulfur proportion (Equation (9)) were calculated as follows:(6)$$SRR=\frac{C_{SO_{4}^{2-}-S,in}-C_{SO_{4}^{2-}-S,out}}{HRT}$$
(7)$$SPR=\frac{C_{TDS-S,out}}{HRT}$$
(8)$$TESPR=\frac{C_{SO_{4}^{2-}-S,in}-C_{S\;species-S,out}}{HRT}$$
(9)$$Elemental\;sulfur\;proportion=\frac{TESPR}{SRR}\cdot 100$$
where $C_{SO_{4}^{2-}-S,in}$ and $C_{SO_{4}^{2-}-S,out}$ are the concentrations of sulfate at the inlet and outlet of the system (See Figure 1), $C_{TDS-S,out}$ is the concentration of total dissolved sulfide in the effluent (outlet), $C_{S\;species-S,out}$ is the sum of all the concentrations of S-species measured in the effluent and *HRT* is the hydraulic retention time of the medium in the middle chamber (volume: 200 mL). The electron recovery for reducing sulfate was calculated as Equation (10):(10)$$\mathrm{Electron}\;\mathrm{recovery}=100\frac{n_{S}\cdot SRR\cdot V\cdot F}{I\cdot M_{S}}$$
where $n_{S}$ is the number of mol of electrons needed to reduce one mole of sulfate to sulfide (= 8 mol e^−^), *SRR* is the sulfate removal rate, *V* is the reactive volume (sum of BES-cathode and FC-anode chamber volumes), *F* is Faraday’s constant (96,485 C mol of electrons^−1^), *I* is the current and $M_{S}$ is the molar weight of sulfur (32.065 g mol^−1^).

## 3. Results and Discussion

### 3.1. Effect of Additional Oxygen Input on the Autotrophic Biocathode

The BES-FC reactor was operated for 161 days under continuous mode and different operational conditions (see Figure 3). Water oxidation took place in the anode of the BES to supply the electrons for the sulfate reduction at the respective cathode. The BES-FC reactor introduced an extra input of oxygen in the reactor during the whole operation (both when the FC was switched on and off). This is because the modified carbon cloth used as air-cathode allowed for the passive diffusion of oxygen into the reactor from the atmosphere, irrespective of whether the FC electrical circuit was open or closed.

The effect of this oxygen permeation through the carbon cloth was evaluated before switching the FC on. During the start-up phase (days 0 to 52, see Figure 1), low electron recoveries for reducing sulfate were observed indicating that there was an additional electron sink in the BES-cathode other than sulfate reduction. This electron flow could be used for: (i) excess H_2_ production, or (ii) reduction of oxygen reaching the biocathode via permeation through the cation-exchange membrane or from the air-cathode side of the FC. The possible excess of oxygen reaching the biocathode chamber could be also used to fully re-oxidize sulfide to sulfate by sulfide oxidizing bacteria such as *Thiobacillus* sp. [40]. This last fact could explain the low sulfate removal rate (SRR) and sulfide production rate (SPR) observed.

Period I started on day 52 achieving a SRR of 646 ± 159 mg SO_4_^2−^-S L^−1^ d^−1^ (Figure 4A). In this period, the SPR was 348 ± 48 mg TDS-S L^−1^ d^−1^ which meant a putative proportion of elemental sulfur conversion of 46.1 ± 13.9% (Figure 4B). At cathode potential of −0.9 V vs. SHE, the sulfate reduction was higher than sulfide oxidation which can be explained by a low passive oxygen diffusion, so the entire sulfide produced was partially oxidized to elemental sulfur. In addition, the electron recovery for reducing sulfate observed was 83.5 ± 19.6% (Figure 4C).

The large standard deviation was potentially caused by different oxygen permeation flows as a result of the carbon cloth becoming covered in salts and/or biofilm [41]. However, the SRR was an order of magnitude higher than the previous investigation of Pozo et al. [42], who achieved 63 mg SO_4_^2−^-S L^−1^ d^−1^ in continuous mode, pH 7.3 and graphite granules as biocathode at the same cathode potential (−0.9 V vs. SHE), and nearly double than Blázquez et al. [21], who achieved 358 mg SO_4_^2−^-S L^−1^ d^−1^ in batch mode without pH control and graphite-fiber brush as biocathode material.

During the whole operation of our BES-FC, the use of a gas diffusion electrode allowed for an extra route of oxygen diffusion into the reactor, which weakened the SRR, but at the same time it improved the theoretical elemental sulfur production rate (TESPR) compared to the results with the BES-EC [32]. As oxygen can be reduced in the BES-cathode to water, the electron recovery is expected to be lower if there is an excess of oxygen. However, the electron recovery in period V was higher than 80%, indicating that the mass transfer of oxygen into the system was minimal, possibly due microorganisms and/or elemental sulfur becoming attached to the carbon cloth cathode [41].

### 3.2. Microbial Electrochemical Cell and Sulfide–Air Fuel Cell Integration

The FC was switched on at day 87 (period II) to evaluate its influence on sulfate treatment, elemental sulfur production and electron recovery. The current density obtained in the FC was 0.09 ± 0.03 A m^−2^ (Figure 4D), which was negligible compared with the current density of the BES (8.2 ± 0.1 A m^−2^) and, thus, the FC was not able to effectively oxidize the sulfide produced by the BES biocathode. The SRR nevertheless increased up to the highest rate observed of 767 ± 26 mg SO_4_^2−^-S L^−1^ d^−1^ and the TESPR up to 386 ± 12 mg S^0^-S L^−1^ d^−1^. The observed low improvement on the TESPR with the FC switched on was a consequence of the low FC current density. This in turn could be due to: (i) a low rate of reaction of TDS with the FC-anode or (ii) an inactivation of the FC electrodes because of elemental sulfur precipitation on the air-cathode surface (partially blocking oxygen diffusion) or by fouling of the catalyst (platinum) by sulfide. To check if the low improvement was due to the low reaction of TDS with the FC-anode, the pH increased up to 7.5 in period III (day 94) to increase the HS^−^ speciation fraction from 50% to 70% (the first proton acidity constant for H_2_S is pKa = 7.0) [43]. Increasing the pH in the cathode has a negative effect on hydrogen evolution (since hydrogen production is less favorable at higher pH values at a fixed potential, according to the Nernst equation) and this resulted in a current decrease. To compensate for this, the biocathode potential was adjusted to −0.95 V vs. SHE. In addition, a CV of the air-cathode was performed to determine if there was an observable reduction of catalytic activity due to precipitation and/or fouling (Figure 2B). The CV showed that after 94 days of operation of the reactor with the FC switched off, the air-cathode was indeed affected, as demonstrated by the much lower current densities attained (Figure 5). For this reason, the air-cathode was replaced at the start of period III.

In period III, the current density of the FC was almost double than in period II (0.17 ± 0.06 A m^−2^), but the SRR decreased to 478 ± 126 mg SO_4_^2−^-S L^−1^ d^−1^ and the SPR and TESPR decreased significantly. As the current density of the BES was almost the same (7.7 ± 1.2 A m^−2^), the decrease on the SRR, SPR and TESPR compared with periods I and II was very likely caused by the new air-cathode that allowed a much higher oxygen diffusion which led to complete sulfide re-oxidation to sulfate again.

The current of the FC was still really low and this poor performance was hypothesized to be caused by the effect of electron-shuttling compounds which are redox mediators that can be repeatedly oxidized and reduced by the electrodes and the microorganisms [44]. To minimize this possible effect, a rubber sheet physically separating the BES-cathode from the FC-anode was installed in period IV (day 145), but keeping both chambers fluidically connected and mixed by separated recirculation. In addition, the current density of the BES was kept constant using galvanostatic mode, controlling the current density at 10 A m^−2^ to avoid fluctuations in the BES performance in period IV. No operational changes resulted in a high SRR improvement and the current density of the FC decreased again to 0.08 ± 0.05 A m^−2^ over time. Figure 4 shows that from mid-period III to the end, the current of the FC tended to decrease probably due to the same inactivation effects previously observed (see Figure 2B).

The FC integration did not improve the system performance as expected mainly because of a deterioration of the air-cathode. The biofilm grown over the air-cathode surface reduced the presence of dissolved oxygen in the bulk liquid and in addition better columbic efficiencies and current production have been observed as in other studies [40,45]. For this reason, the biofilm attached to the air-cathode should not decline the FC performance. The poor performance of the FC might be caused by excess of oxygen reacting with sulfide, by the elemental sulfur attached to the air-cathode or by inactivation of the air-cathode platinum catalyst as observed in other metal-based catalysts [22].

A membrane next to the air-cathode could be a solution for this problem, which was also studied by Zhao et al. [27]. Other studies tried to remove sulfate heterotrophically and then oxidize the sulfide in the anode of a single-chamber reactor, achieving proportions of elemental sulfur between 60 and 75% of the initial sulfate [28,29] but at lower SRR and at expenses of an external electron donor supply. Further studies should test the same system but without the FC, while still allowing oxygen intrusion through a gas diffusion layer. Additionally, the current configuration should be trialed with an ion-exchange membrane to replace the mesh separator between FC-anode and cathode, as a means to prevent direct oxidation of sulfide by oxygen.

In comparison to our previous study [32], the FC improved significantly the proportion of elemental sulfur but with lower SRR and electron recoveries. However, in terms of energy consumption for sulfur recovery, the BES-FC shows an improvement compared with the BES-EC that reported an energy consumption per kg of sulfate removed of 9.18 ± 0.80 kWh kg^−1^ SO_4_^2−^-S considering the maximum SRR. Converting this result into kg of sulfur recovered and adding the energy consumption of the EC, this means an energy consumption of 18.74 ± 1.81 kWh kg^−1^ S^0^-S. With our BES-FC, the energy consumption for sulfate removed was slightly higher (10.61 ± 0.12 kWh kg^−1^ SO_4_^2−^-S) because, as mentioned before, the extra oxygen permeation caused an inherent inefficiency (re-oxidation of some of the sulfide to sulfate). However, as the FC was not consuming energy, the total energy consumption for sulfur production was of 16.50 ± 0.19 kWh kg^−1^ S^0^-S, a 12% improvement in terms of energy consumption for elemental sulfur production than the use of a BES-EC and at lower capital investment needs (although the savings can only be determined in larger-scale trials), since the FC does not require a potentiostatic control/power source.

## 4. Conclusions

This work demonstrates a BES-FC that can treat synthetic wastewater with high sulfate content obtaining sulfate removal rates up to 767 mg SO_4_^2−^-S L^−1^ d^−1^ at −0.9 V vs. SHE. In addition, the BES-FC allowed for a 12% reduction in the energy consumption per kg of elemental sulfur recovered compared with a BES with an electrochemical cell instead of the FC with an air-cathode. The FC improved the proportion of elemental sulfur produced thanks to the oxygen diffusion through the carbon cloth compared with an electrolysis cell, but at expenses of lower sulfate removals.

## Figures and Tables

**Figure 1 ijerph-18-05571-f001:**
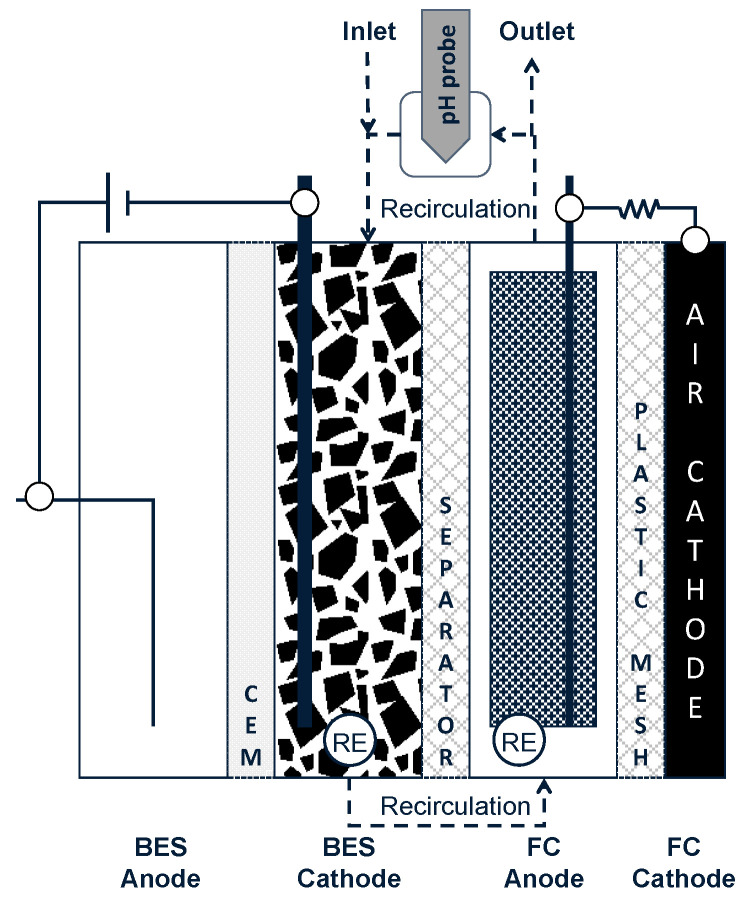
Schematic of BES-FC setup. The separator consisted of a plastic mesh (5 × 5 mm^2^ of grid and 1 mm of thickness). It was replaced by rubber after 145 days of operation. CEM: cation-exchange membrane, RE: reference electrode.

**Figure 2 ijerph-18-05571-f002:**
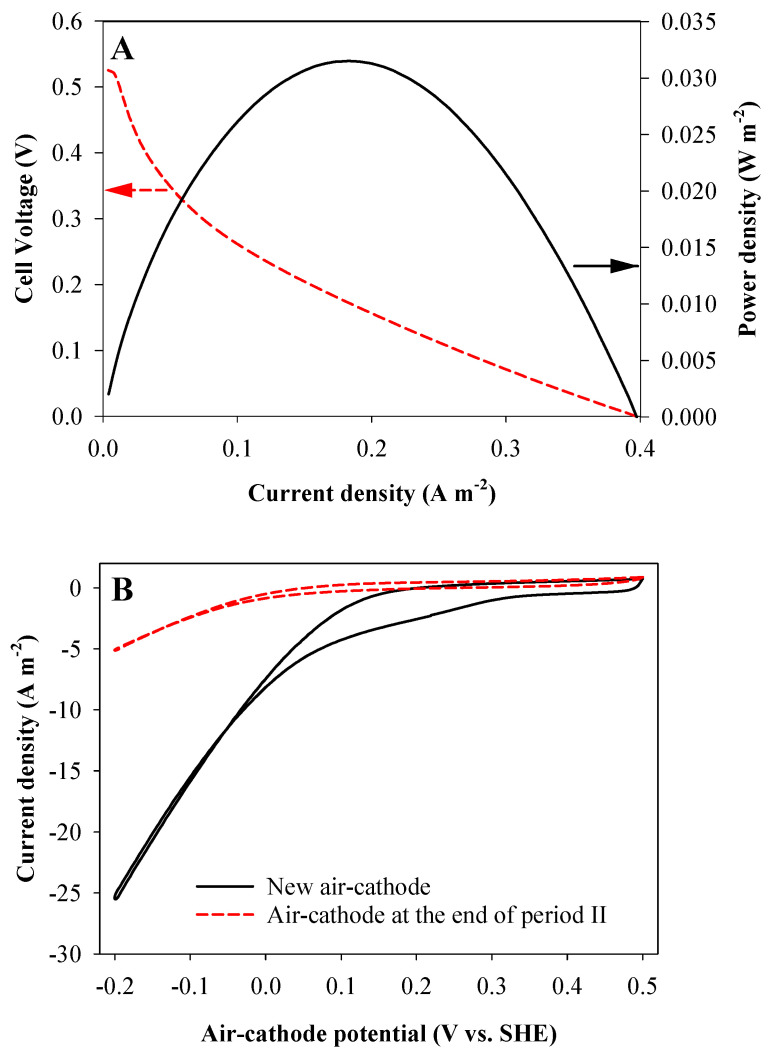
Electrochemical techniques for the characterization of the FC; (**A**): polarization curve of the FC and (**B**): cyclic voltammetry of the air-cathode at the end of period II and of a new piece of air-cathode. Scan rate of both analyses: 1 mV s^−1^.

**Figure 3 ijerph-18-05571-f003:**
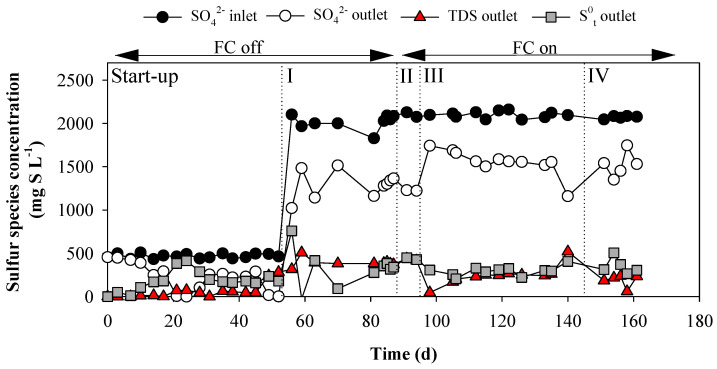
Evolution of sulfur species concentration along the BES-FC operation phases, sulfate in the inlet and outlet, total dissolved sulfide (TDS) in the outlet and theoretical elemental sulfur produced (T-S^0^) in the outlet. N.B. the lack of statistically significant discrepancy between TDS and S^0^ at the outlet indicate that there is no formation of intermediate sulfur products, as previously shown by Pozo et al. [30].

**Figure 4 ijerph-18-05571-f004:**
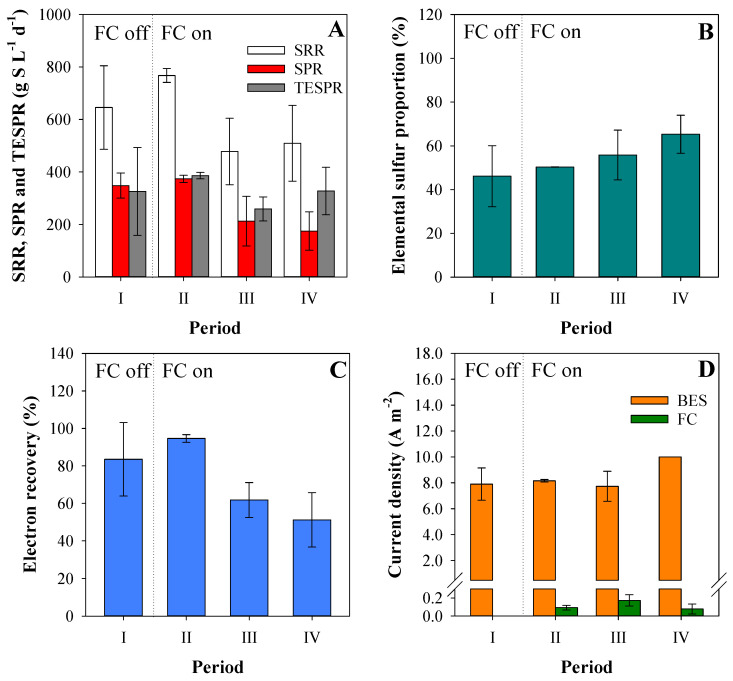
Average plots of the different operational phases of the BES-FC after the start-up on (**A**): sulfate reduction rate (SRR), sulfide production rate (SPR) and theoretical elemental sulfur production rate (TESPR), (**B**): elemental sulfur proportion in relation to sulfate reduced, (**C**): electron recovery as sulfate reduced in the biocathode and (**D**): current density of BES and FC.

**Figure 5 ijerph-18-05571-f005:**
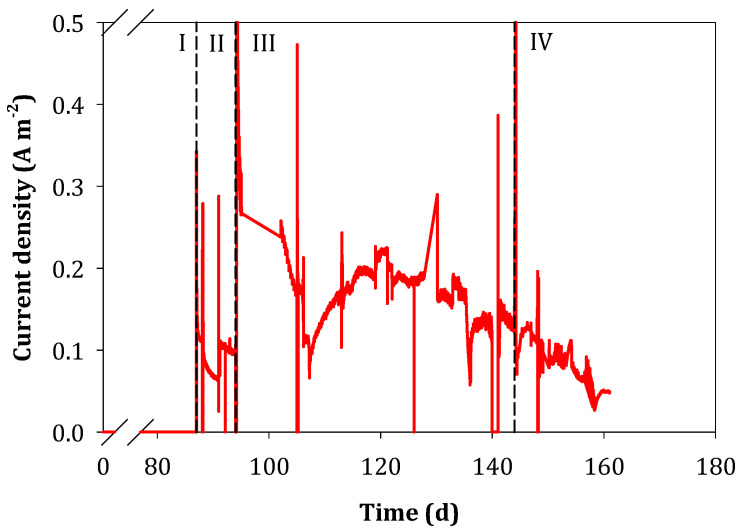
Current density of the FC during all the experimental phases. N.B. The start-up and period I are condensed because the FC was in open circuit during this time (no current).

**Table 1 ijerph-18-05571-t001:** Operational conditions in each experimental period.

Period	Days of Operation	Sulfate Influent [mg S L^−1^]	BES-Cathode Potential [V vs. SHE]	pH	HRT [d]	FC Operation
Start-up	0–52	500	−0.7 to −0.9	7.0	3.1–1.1	Off
I	52–87	2000	−0.9	7.0	1.1	Off
II	87–94	2000	−0.9	7.0	1.1	On
III ^1^	94–145	2000	−0.95	7.5	1.1	On
IV ^2^	145–161	2000	−0.98 ± 0.01	7.5	1.1	On

^1^ Air-cathode of the FC replaced.^2^ A non-porous rubber separator was added as physical separation between biocathode and anode, although both chambers remained fluidically connected through the recirculation line (Figure 1). The current density of the BES was controlled at 10 A m^−2^.

## Data Availability

Data are contained within the article.

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
