# Peer review of "Implementation of a Sulfide–Air Fuel Cell Coupled to a Sulfate-Reducing Biocathode for Elemental Sulfur Recovery"

_ijerph, 2021, doi:10.3390/ijerph18115571_

Round 1
Reviewer 1 Report
This paper deals with a novel and flexible bio-electrochemical systems (BES) for treating several types of wastewaters and recovering valuable products concomitantly by coagulants thatcombine bio-electrochemical sulfate reduction. The sulfate was removed from processes of synthetic wastewater with high sulfate content via the BES-FC were investigated by several characterization techniques. The studies were quite systematic and the resulted were well organized by the authors. I’d like to recommend the publication of this paper in International Journal of Environmental Research and Public Health after revision.
- The author should explain why pH value should be kept at 7.0 and 7.5 in the operational conditions.
- The author should provide the durability of bioelectrochemical systems- sulfide-air fuel cell by using the same anode and cathode. In addition, the condition of anode and cathode should be observed by using XRD and FE-SEM/EDX.
- The author should explain whether the intermediate product form in the process.
- The author should provide more details of the mechanism in reactions of processes
Reviewer 2 Report
All the comments are in the document attached.
